# α-Actinin-4 recruits Shp2 into focal adhesions to potentiate ROCK2 activation in podocytes

Chien-Chun Tseng[1],*, Ru-Hsuan Zheng[1],*, Ting-Wei Lin[1], Chih-Chiang Chou[1], Yu-Chia Shih[1], Shao-Wei Liang[1], Hsiao-Hui Lee[1,2]

**Cell–matrix adhesions are mainly provided by integrin-mediated focal adhesions (FAs). We previously found that Shp2 is essential for FA maturation by promoting ROCK2 activation at FAs. In this study, we further delineated the role of α-actinin-4 in the FA recruitment and activation of Shp2. We used the conditional immortalized mouse podocytes to examine the role of α-actinin-4 in the regulation of Shp2 and ROCK2 signaling. After the induction of podocyte differentiation, Shp2 and ROCK2 were strongly activated, concomitant with the formation of matured FAs, stress fibers, and interdigitating intracellular junctions in a ROCK-dependent manner. Gene knockout of α-actinin-4 abolished the Shp2 activation and subsequently reduced matured FAs in podocytes. We also demonstrated that gene knockout of ROCK2 impaired the generation of contractility and interdigitating intercellular junctions. Our results reveal the role of α-actinin-4 in the recruitment of Shp2 at FAs to potentiate ROCK2 activation for the maintenance of cellular contractility and cytoskeletal architecture in the cultured podocytes.**

## Introduction

Adherent cells are constantly exposed to the mechanical stimuli arising from the surrounding environment, such as ECM. By the process called mechano-transduction, the extracellular mechanical stimuli are sensed and converted into the intracellular biochemical and electrical signals that elicit various adaptive cellular responses (Discher et al, 2005; Jaalouk & Lammerding, 2009; Wang et al, 2009; Wozniak & Chen, 2009; Kolahi & Mofrad, 2010). Cells interact with ECM by the adhesion receptors, such as integrins. The activation of integrins by ECM engagement facilitates the recruitment of multiple proteins to form nascent adhesions. Thereafter, some adhesions turnover rapidly, and some adhesions become matured focal adhesions (FAs) in a force-dependent manner (Schwartz &

DeSimone, 2008; Geiger et al, 2009). The matured FAs connect with the contractile actin cytoskeleton and transmit mechanical forces between the ECM and the cytoskeleton. The organization of cell adhesion machinery involves several networks connecting the sensory and the operational modules, as well as the interplay between the cytoskeletal machinery; and FA is a key link for cells to adapt their physical conditions to those of the surrounding environment. The small GTPase RhoA is one of the key regulators of actin cytoskeleton rearrangement, which functions via its effector Rho-associated protein kinase (ROCK). Activation of RhoA/ROCK signaling promotes the phosphorylation of myosin light chain (MLC) and results in the force generation (McBeath et al, 2004; Connelly et al, 2010). These forces activate some mechanosensitive events that recruit additional cytoskeletal and signaling proteins at FAs, to generate a positive feedback mechanism for cytoskeleton rearrangement and FA strengthening (Webb et al, 2002; Zaidel-Bar et al, 2007; del Rio et al, 2009; Huveneers & Danen, 2009; Parsons et al, 2010).

In our previous studies, we have found that ROCK2 can be negatively regulated by phosphorylation at the Y722 residue (Lee & Chang, 2008). This modification, which reduces its interaction with RhoA, is regulated reciprocally by c-Src kinase and Shp2 phosphatase at FAs (Lee & Chang, 2008; Lee et al, 2010). We also demonstrated that Shp2 plays an important role in the regulation of intracellular tension in cells response to matrix rigidity through its effects on FA maturation and stress fiber orientation in MEFs (Lee et al, 2013). Shp2 is encoded by the PTPN11 gene (Tartaglia et al, 2001). It is a ubiquitously expressed nonreceptor protein tyrosine phosphatase (PTP), characterized by having two Src homology-2 (SH2) domains in the N-terminal region that autoinhibits its PTP catalytic function (Neel et al, 2003). Because Shp2 is a cytosol protein, the regulation of Shp2 recruitment at FAs provides a spatiotemporal control of RhoA-mediated ROCK2 activation for FA maturation. However, the molecular mechanism of how Shp2 is recruited at FAs is still unknown. In this study, we identified α-actinin-4 as a Shp2-interacting protein at FAs. We also demonstrated that α-actinin-4 facilitates Shp2 activation in MEFs. Based on our results, we hypothesize that α-actinin-4 assists Shp2

[1]Department of Life Sciences and Institute of Genome Sciences, National Yang Ming Chiao Tung University, Taipei, Taiwan    [2]Center for Intelligent Drug Systems and Smart Bio-Devices (IDS2B), National Yang Ming Chiao Tung University, Taipei, Taiwan

Correspondence: hhl@nycu.edu.tw
*Chien-Chun Tseng and Ru-Hsuan Zheng contributed equally to this work.

recruitment at FAs and then subsequently enhances ROCK2-mediated contractility for FA maturation.

α-Actinins belong to the spectrin family of actin cross-linking proteins. In mammals, there are four genes that encode the highly homologous forms of α-actinin proteins, *ACTN1* to 4. *ACTN1* and *ACTN4* are ubiquitously expressed and known to be localized at FAs; *ACTN2* and *ACTN3* are expressed specifically in muscle cells (Otey et al, 1990; Sjoblom et al, 2008; Oikonomou et al, 2011; Foley & Young, 2014). Human genetic mutations of *ACTN4* cause an autosomal-dominant focal segmental glomerulosclerosis (FSGS) (Kaplan et al, 2000), a kidney lesion characterized by alteration in podocytes (Mundel & Shankland, 2002; Lowik et al, 2009). *Actn4* knockout mice develop severe proteinuria and eventually suffer from kidney failure (Kos et al, 2003). Podocytes generated from the ACTN4$^{-/-}$ mice were markedly less adherent when subjected to shear stress in comparison with those subjected to reintroducing of α-actinin-4 (Dandapani et al, 2007), suggesting the crucial role of α-actinin-4 for podocyte adhesion strengthening. Podocytes are highly specialized visceral epithelial cells lining the outer surface of the glomerular capillaries. They have a complex cellular architecture consisting of cell body and major processes (MPs) that extend outward from their cell body, forming interdigitated foot processes (FPs) that enwrap the glomerular capillaries (Sachs & Sonnenberg, 2013; Lal et al, 2015). FPs are actin-rich projections anchored to the glomerular basement membrane via focal adhesion (Scott & Quaggin, 2015). The function of podocyte relies on the maintenance of cell attachment and their unique architecture in the adaptation to mechanical stress caused by the fluid filtration. The mechano-biology of the regulation of cell adhesion and contractile force has been one of the most critical issues in the physiological and pathological implications of podocyte biology.

In this study, we used a mouse temperature-inducible podocyte line to verify our hypothesis of the role of α-actinin-4 in the regulation of Shp2 and ROCK2 activation underlying adhesion signaling. We found that the Shp2 and ROCK2 activations were significantly up-regulated after podocytes were induced to differentiation in vitro. Gene knockout of *Actn4* abolished Shp2 and ROCK2 activation and reduced FA maturation in podocytes, as well as a decrement of matured FAs in *Actn4*$^{-/-}$ podocytes. We also demonstrated that the loss of ROCK2 impaired the cellular contractility and architecture of podocytes. In summary, our findings provide a new insight into the role of α-actinin-4 at FAs, by showing that α-actinin-4 participates in the recruitment of Shp2 at FAs to promote ROCK2 activation, which in turn contributes to enhance actomyosin contractility and cell adhesion. This mechanism underpins the maintenance of cellular architecture and function in podocytes.

# Results

### Identification of α-actinin-4 as a Shp2-interacting protein at focal adhesions

We have previously demonstrated that Shp2 serves as a key regulator at FAs to promote RhoA-mediated ROCK2 activation in cells response to matrix rigidity (Lee & Chang, 2008; Lee et al, 2013). We

also found that the Shp2 N-SH2 domain, a phosphotyrosine binding domain, is required for Shp2 targeting at FAs in MEFs (Lee et al, 2013). Here, we further observed that the level of Shp2 in the FA fraction was significantly lower when cells were treated with FAK kinase inhibitor (Figs 1A and S1), suggesting that Shp2 targeting at FAs could be regulated by FAK-mediated signaling. To search for the FAK-regulated Shp2-interacting proteins in FAs, FA fractions isolated from MEFs treated with or without FAK inhibitor were incubated with His-tagged Shp2 N-SH2 domain recombinant protein and then pull-down with Ni-beads, SDS–PAGE, and silver staining. The levels of pulled-down materials, changed by FAK inhibition, were identified by mass spectrometry analysis (Fig 1B and Table S1). One of them, one major band with a molecular weight of about 100 kD, was identified mainly as α-actinin-4; in addition, α-actinin-1 was also detected. We used anti-Shp2 antibody to immunoprecipitate the endogenous Shp2 protein from the isolated FA fraction and found that α-actinin-4, but not α-actinin-1, was co-immunoprecipitated (Fig 1C and D). We also demonstrated that the transient expressed GFP-α-actinin-4, but not mCherry-α-actinin-1, was co-immunoprecipitated with flag-Shp2 in HEK-293T cells (Fig S2A and B). Furthermore, we also found (via anti-phosphotyrosine antibody 4G10) that GFP-α-actinin-4 was tyrosine phosphorylated (Fig S2C). To know whether the interaction between α-actinin-4 and Shp2 N-SH2 requires tyrosine phosphorylation, FA fractions was treated with λPPase; we found that treatment of λPPase significant diminished the levels of α-actinin-4 pulled-down by N-SH2 (Fig 1E). All these data suggest that α-actinin-4 could specifically interact with Shp2 in a phosphotyrosine-dependent manner.

To confirm this interaction, MEFs were seeded on fibronectin (FN)-coated glass coverslips and fixed for the in situ proximity ligation assay (PLA) with anti-Shp2 and anti-α-actinin-4 antibodies. The F-actin was further stained with phalloidin. The signal of PLA assay was observed at the termini of stress fibers (Fig 2A), whereas the PLA signal was undetectable in the *Ptpn11*$^{Ex3-/-}$ MEFs, where exon3 deletion disrupted the N-SH2 domain of Shp2, leading to the loss of FA-targeting ability. In addition, the PLA reaction without the anti-α-actinin-4 antibody in wild-type MEFs showed a clean PLA background, suggesting the specificity of the detected PLA signal (Fig 2A). We further analyzed the effect of FAK inhibition on this interaction; as expected, the PLA signal of interaction between Shp2-and α-actinin-4 was significantly reduced by FAK inhibition (Fig 2B). Taken together, our data suggest that Shp2 may interact with α-actinin-4 through its N-SH2 domain at FAs in a FAK kinase–dependent manner.

### α-Actinin-4 facilitates Shp2 FA targeting and activation in MEFs

To study the role of α-actinin-4 in the regulation of Shp2, two clones of *Actn4*$^{-/-}$ MEFs were generated by the CRISRP/Cas9 method (Fig 3A). We observed that the levels of Shp2 protein in the isolated FA fraction were reduced in both *Actn4*$^{-/-}$ clones compared with the parental wild-type cells (Fig S3), suggesting that α-actinin-4 might play a positive role in the regulation of Shp2 FA targeting. We then used a FRET-based Shp2 biosensor, Shp2-SWAP, to examine the Shp2 activation level in living cells (Sun et al, 2013). The Shp2 activation level was reflected by the FRET efficiency, which was significantly higher when cells were attached on FN compared with

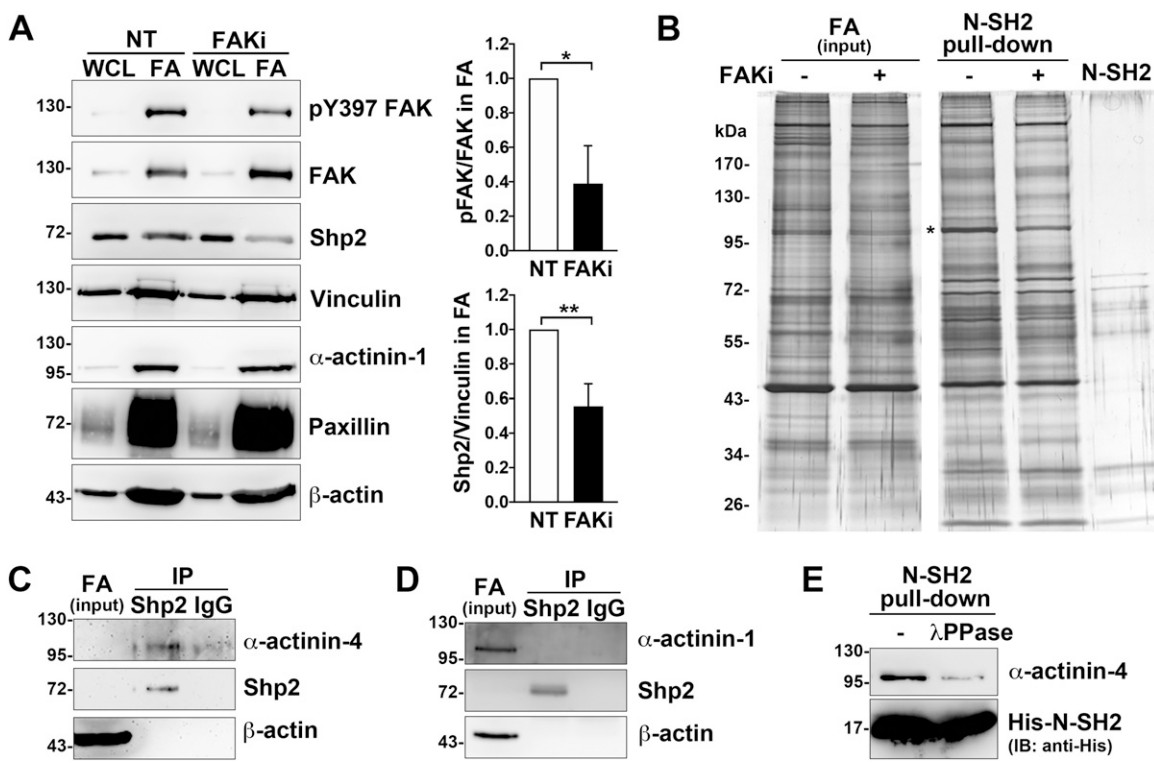

**Figure 1.  Identification of Shp2-interacting proteins in FAs.**
MEFs seeded on FN-coated dishes were treated with or without FAK inhibitor 14 (15 μM) for 90 min for isolation of FAs as described in the Materials and Methods section. **(A)** Whole-cell lysate (WCL; 1%) and isolated FA fractions (FA) were subjected to Western analysis as indicated. Relative FAK Y397 phosphorylation and levels of Shp2 versus vinculin in FA fractions from four independent experiments were measured. **(B)** FA fractions were incubated with 50 μg of His-tagged Shp2 N-SH2 recombinant protein followed by pulled-down with Ni-beads, SDS–PAGE, and sliver staining. By mass spectrometry analysis, the band indicated by * was identified as α-actinin-4 and α-actinin-1. **(C, D)** FA fractions isolated from non-treated MEFs were immunoprecipitated with anti-Shp2 antibody followed by Western blotting analysis as indicated. The concentration of Shp2 and α-actinin-4 in the IP input sample might be too low to be detected here. **(E)** FA fractions were incubated with or without 400 U of λPPase for 30 min before pull-down with His-N-SH2 proteins and Ni-beads. The levels of pulled-down α-actinin-4 was detected by Western blotting as indicated. Data are mean ± SD. *P < 0.05, **P < 0.01 (two-tailed, paired t test).
Source data are available for this figure.

cells attached to poly–l-lysine (PLL) (Fig S4). Wild-type and _Actn4_⁻/⁻ MEFs were transiently transfected with the expression construct of Shp2-SWAP and replated on the FN-coated glass bottom dishes for FRET imaging. Our results show that the Shp2 activation level was significantly reduced in both two clones of _Actn4_⁻/⁻ MEFs compared with the WT MEFs (Fig 3B and D). Reintroducing the expression construct of mApple-α-actinin-4 in both two _Actn4_⁻/⁻ clones rescued the Shp2 activation level (Fig 3C and D), suggesting the positive role of α-actinin-4 in the regulation of Shp2 activation. Based on these results and the finding in our previous studies, we hypothesized that α-actinin-4 may contribute to the recruitment and activation of Shp2 at focal adhesions and subsequently enhances ROCK2 signaling to strengthen cell adhesions.

## The contractile signaling mediated by Shp2 and ROCK2 in cultured podocytes

It has been known that the interaction between α-actinin-4 and integrin adhesion receptors is critical for maintaining strong podocyte adhesion to the glomerular basement membrane (Dandapani et al, 2007). We used a conditional immortalized mouse podocyte cell line to verify our proposed molecular mechanism of

α-actinin-4 in the regulation of Shp2/ROCK2 signaling at FAs. Cells were maintained for growth in the medium in the presence of INFγ at 33°C for subsequent immortalization by a temperature-sensitive variant of the SV40 large T antigen under the control of INFγ-inducible promoter (Schiwek et al, 2004) and were induced to differentiate by temperature-switch to the nonpermissive 37°C in the absence of INFγ for 10–14 d. The differentiation status of the podocytes was confirmed by the expression of mature podocyte marker synaptopodin (Mundel et al, 1997). By Western blotting, the expression of synaptopodin was significantly detected in cells after temperature-switch, indicating these podocytes had been differentiated (Fig 4A). The calculated molecular weight of synaptopodin renal isoform, encoded by gene _SYNPO_, is about 96 kD (Ning et al, 2020). In our results, the synaptopodin band was detected at ~130 kD, which is consistent with other reports showing a higher than expected mass, possibly because of posttranslational modification (Asanuma et al, 2005; Ning et al, 2020). The podocyte-specific protein podocin was also detected in the cells under both conditions (Fig 4A). By immunofluorescence (IF) staining with anti-paxillin antibody and phalloidin, significant FAs and stress fibers were observed in the differentiated podocytes (Fig 4B). Notably, the

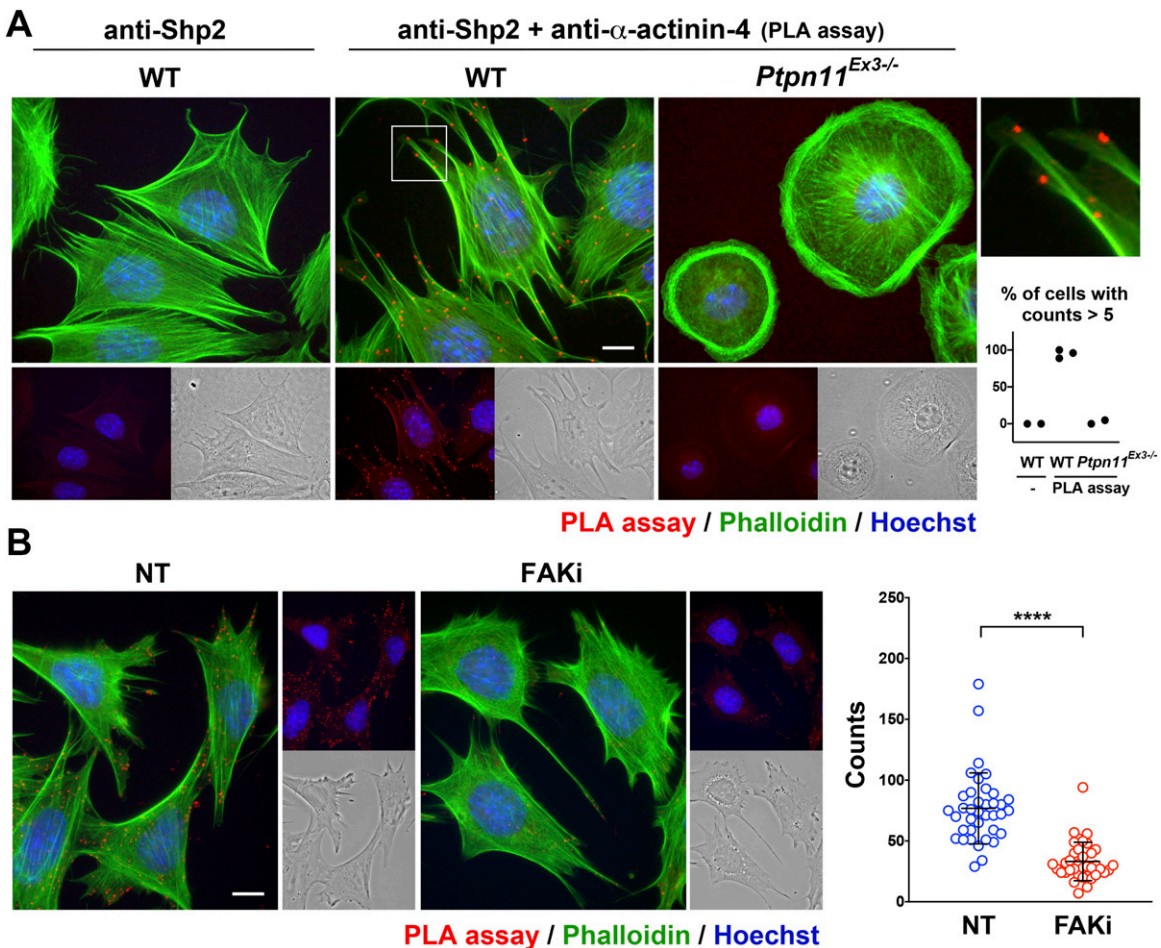

**Figure 2. Shp2 interacts with α-actinin-4 at FAs.**
**(A)** Wild-type (WT) and *Ptpn11*^Ex3-/-^ MEFs were seeded on FN-coated coverslips for 2 h and fixed for in situ proximity ligation assay (PLA) with anti-Shp2 plus anti-α-actinin-4 antibodies (red). Reaction with anti-Shp2 antibody serves as a negative control. After reaction, cells were stained with FITC-phalloidin (green) and Hoechst (blue) for F-actin and nucleus, respectively. Scatter dot plots show the percentage of cells with PLA count >5 in each independent experiments. **(B)** MEFs seeded FN-coated coverslips were treated with or without FAK inhibitor 14 (20 μM) for PLA assay. Scatter dot plots (mean ± SD) show PLA counts from more than 36 cells in three independent experiments; each dot represents one single cell. ****$P < 0.0001$ (Mann–Whitney U test). Scale bars, 10 μm.

specialized interdigitating intercellular junctions, which is a unique intercellular junction structure in cultured podocytes, was also observed in the differentiated podocytes (Fig 4B).

The activation of Shp2, indicated by Shp2 Y542 phosphory-lation (Neel et al, 2003), was strongly enhanced in the differentiated podocytes (Fig 4C). The phosphorylation of FAK at Y397 and paxillin at Y31 was also increased in the differentiated podocytes compared with the non-differentiated cells, suggesting the activation of FAK-mediated signaling (Fig 4C). We also found that activation of ROCK2 indicated by it autophosphor-ylation at S1366 (Chuang et al, 2012) was significantly enhanced in the differentiated podocytes; whereas the activation of ROCK1, indicated by S1333 phosphorylation (Chuang et al, 2013), was insignificant (Fig 4D). The phosphorylation level of ROCK downstream substrate MLC was also enhanced, suggesting the increase of cellular contractility in the differentiated podocytes (Fig 4D). In addition, treatment of cells with ROCK inhibitor Y27632 dramatically reduced the phosphorylation of ROCK2 activation and

MLC in the differentiated podocytes (Fig 4E). As expected, the for-mation of FAs, stress fibers, and interdigitating intercellular junctions were also diminished (Fig 4F and G), suggesting the critical role of ROCK-mediated contractility for the maintenance of cellular ar-chitecture of cultured podocytes. Although cultured podocytes do not have convincingly convincing primary and tertiary foot processes to form the podocyte-specific slit diaphragms, the significant activation of Shp2 and ROCK2 detected in the cells after temperature-switch suggests that it could be a feasible cell model to verify our hypothesis.

### α-Actinin-4 is crucial for adhesion-mediated Shp2 activation in cultured podocytes

To examine the role of α-actinin-4 for Shp2 activation in cultured podocytes, two *Actn4*^-/-^ podocyte clones were generated by the CRISRP/Cas9 method (Fig 5A). The induction of synaptopodin expression in both *Actn4*^-/-^ podocyte clones after temperature-

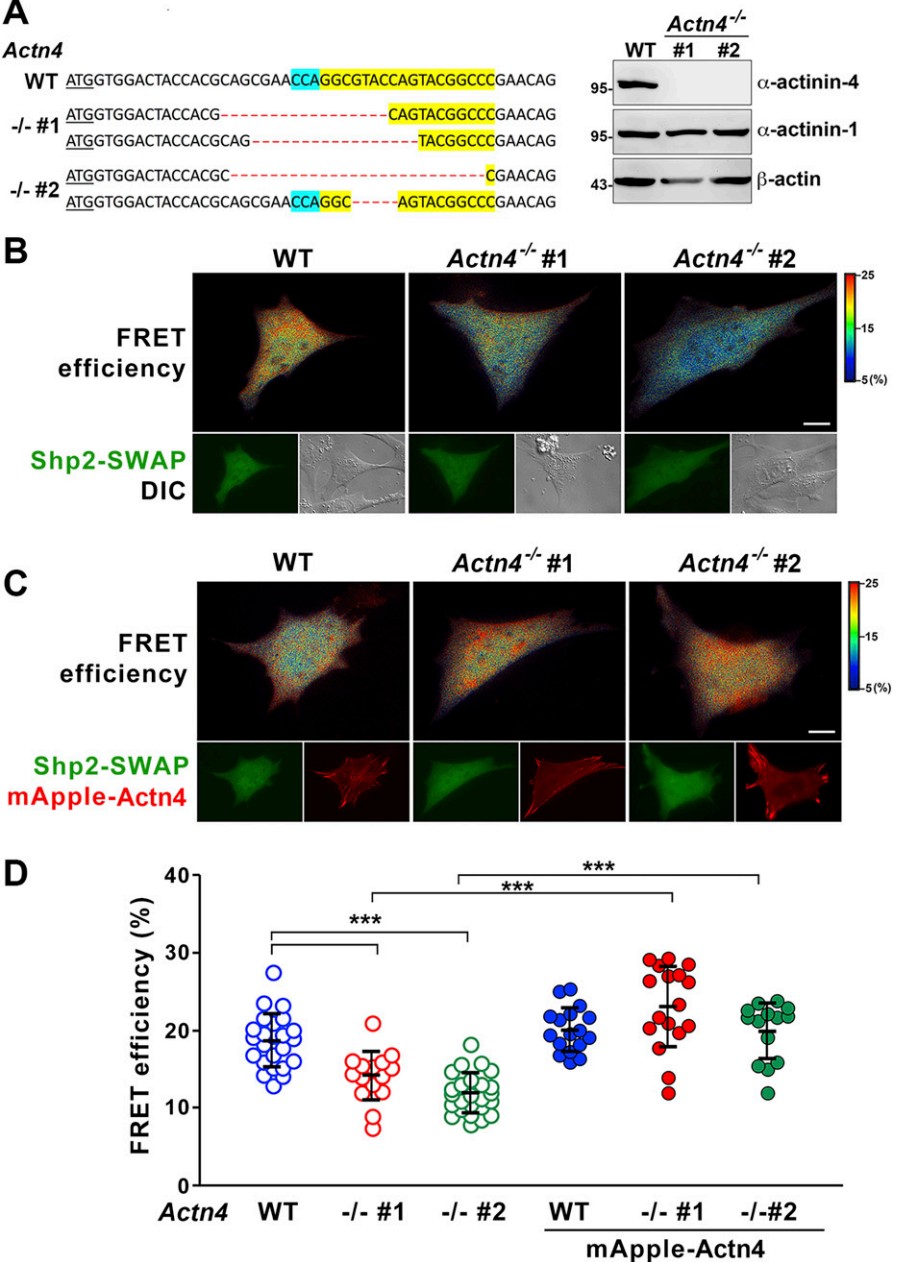

**Figure 3. Loss of α-actinin-4 reduces Shp2 activation.**
**(A)** The modified DNA sequence of Actn4 in two selected Actn4$^{-/-}$ MEF clones. The gRNA targeted region is highlighted in yellow; PAN site is shown in light blue, and modified DNA sequence is shown in red. The expression of α-actinin-4 in these clones was checked by Western blotting. **(B)** The FRET efficiency images of a wild-type or Actn4$^{-/-}$ MEF transfected with Shp2 FRET reporter (Shp2-SWAP). **(C)** The expression construct of mApple-Actn4 (red) was co-transfected with Shp2-SWAP into MEF cells for FRET imaging analysis. **(D)** Scatter dot plots (mean ± SD) of the Shp2-SWAP FRET efficiency. The sample numbers are 24 and 17 for WT 16 and 18 for Actn4$^{-/-}$ #1 and 25 and 15 and 24 for Actn4$^{-/-}$ #2 cells in the absent or present of mApple-Actn4 co-transfection, respectively. Differences between continuous variables were compared using the Mann–Whitney U test. ***P < 0.0005. Scale bars, 10 μm.
Source data are available for this figure.

switch suggests that loss of *Actn4* did not affect the podocyte differentiation in vitro (Fig 5A). Via PLA assay, the interaction between Shp2 and α-actinin-4 was significantly detected in differentiated podocytes (Fig S5A). We then compared the levels of Shp2 pY542 phosphorylation, which indicated its activation in these podocyte clones, and found that the induction of Shp2 Y542 phosphorylation signal in differentiation podocytes was diminished in both *Actn4*$^{-/-}$ podocyte clones compared with the wild-type parental clone (Fig 5B). Interestingly, the Shp2 activation signal in differentiated podocytes was abolished when podocytes were kept in suspension for 30 min (Fig 5C). The activation level of ROCK2, indicated by S1366 phosphorylation, was consistent with Shp2 activation signal. Thus, the α-actinin-4 regulated Shp2 activation and the subsequent ROCK2 activation were both mediated on the basis of cell adhesion signaling. We measured the number and the area of FA in these podocytes and found that the number of the matured FA (area >1 μm$^2$) significantly decreased in both *Actn4*$^{-/-}$ clones, compared with that in wild-type podocytes; whereas the number of small FA (area < 1 μm$^2$) showed no significant difference (Fig 5D). The formation of interdigitating intercellular junctions appeared to be slightly affected by the loss of α-actinin-4 (Fig S5B). Taken together, our data suggest that α-actinin-4 is crucial in the regulation of Shp2 activation to strengthen cell adhesion through its effect on enhancing ROCK2 signaling for FA maturation in cultured podocytes.

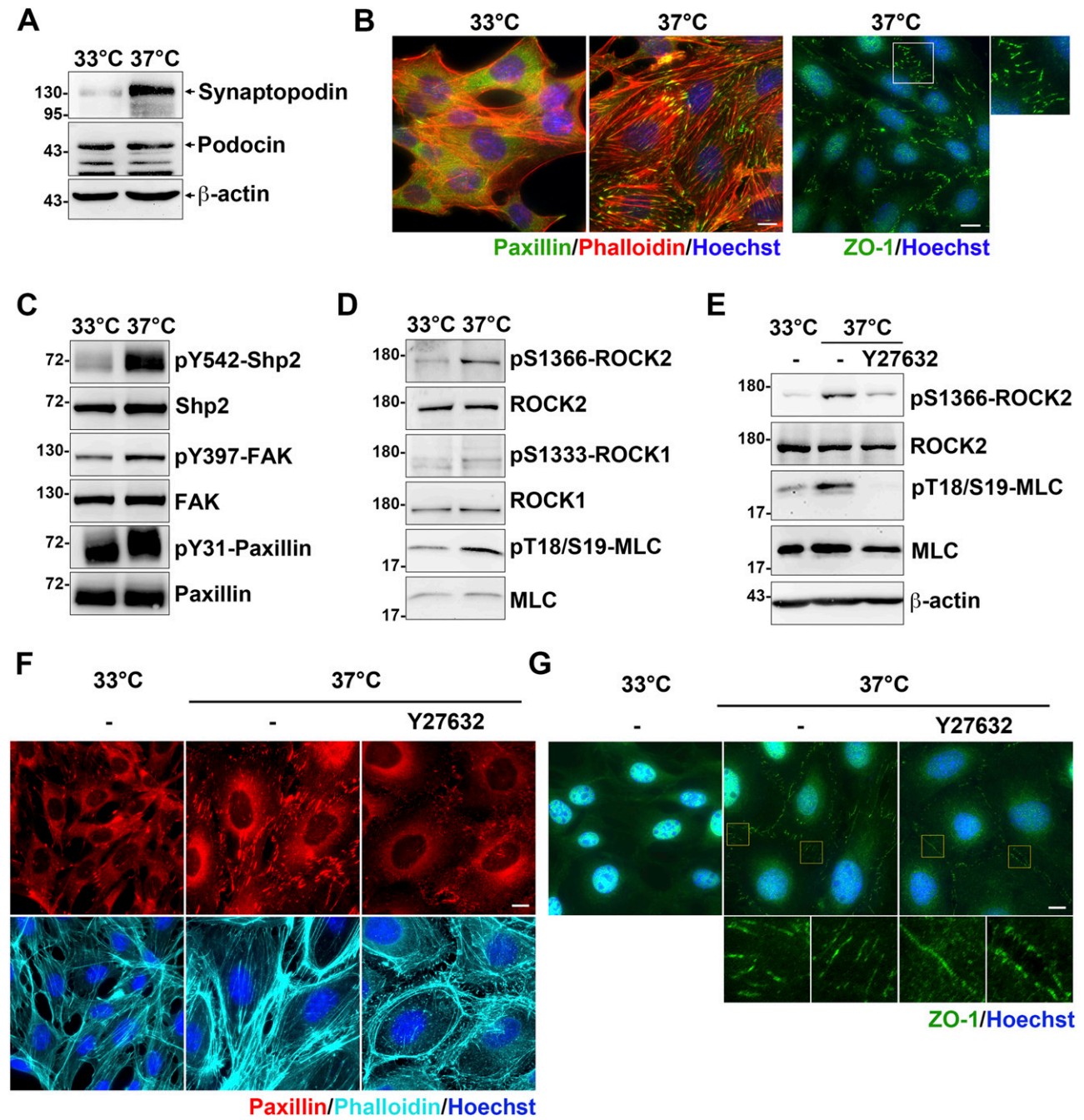

**Figure 4.   The contractile signals in cultured podocytes.**
The conditional immortalized mouse podocytes cultured at 33°C in the present of INFγ (proliferative) or at 37°C in the absent of INFγ (differentiated) for 10–14 d on the type-IV collagen–coated substrates. **(A)** Podocytes were harvested for Western blot analysis with antibodies specific for podocyte marker proteins. **(B)** Podocytes seeded on type-IV collagen–coated glass coverslips were fixed for immunofluorescence staining with anti-paxillin and anti-ZO-1 antibodies for detecting FA and intercellular junctions and phalloidin and Hoechst for F-actin and DNA, respectively. **(C, D)** Protein extracts form podocytes were subjected to Western blot analysis with antibodies as indicated. **(E, F, G)** Podocytes were serum-starved for 24 h and then treated with or without 20 μM of Y27632 for 1 h. Cells were harvested for Western blotting analysis or fixed for immunofluorescence staining as indicated. Scale bars, 10 μm.
Source data are available for this figure.

## ROCK2 is essential for cellular contractility in cultured podocytes

To know the effects of Shp2 on ROCK2 activation, the differentiated podocytes were treated with Shp2 inhibitor IIB-08. We found that ROCK2

S1366 phosphorylation decreased as Shp2 Y542 phosphorylation was reduced (Fig S6), suggesting the crucial role of Shp2 for ROCK2 activation in cultured podocytes. Because there are two isoforms of ROCKs, ROCK1, and ROCK2, in mammals, we then generated two *Rock2*⁻/⁻ podocyte

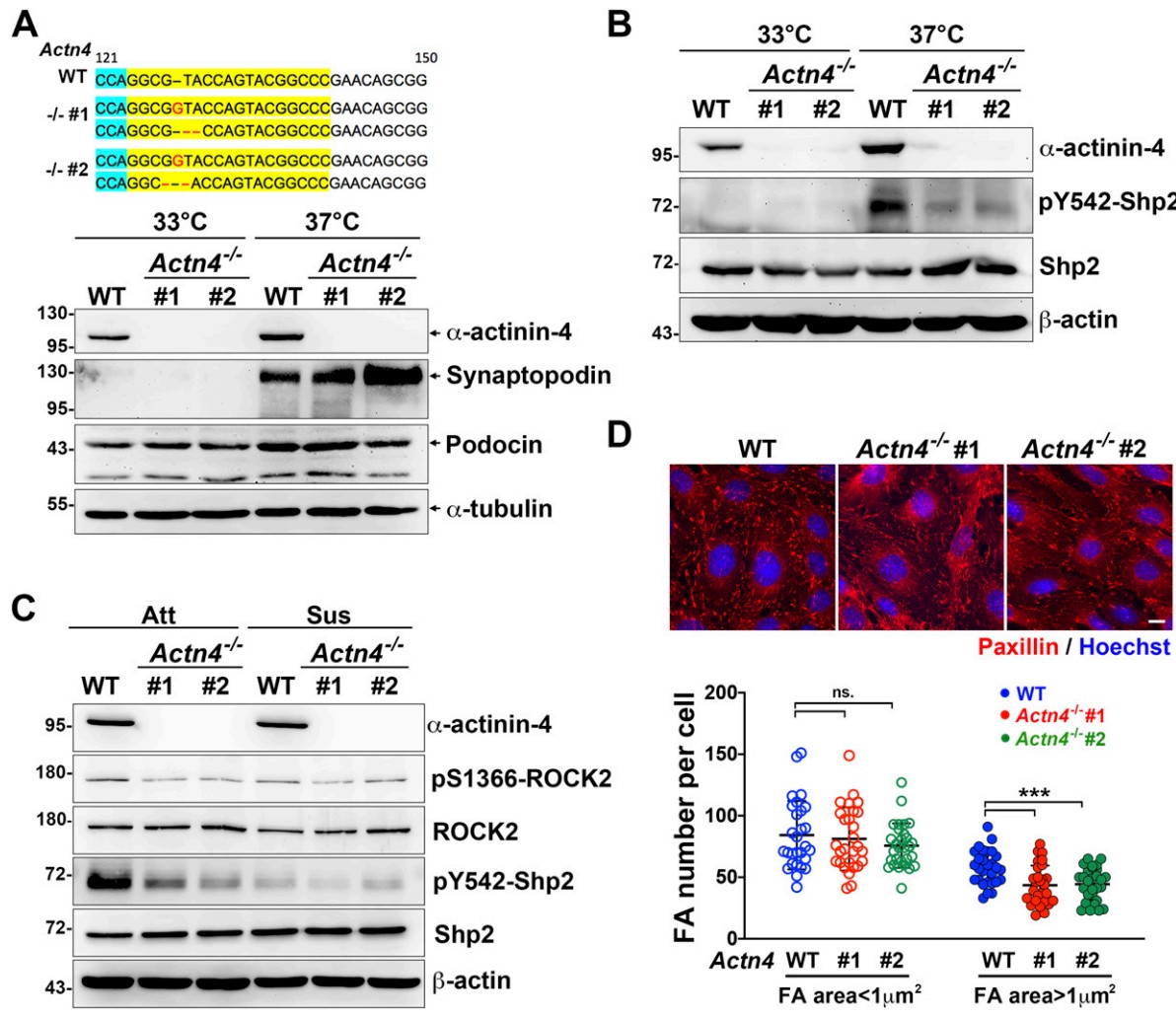

**Figure 5.   α-Actinin-4 is required for Shp2 activation underlying adhesion signaling in podocytes.**
**(A)** The modified DNA sequence of two selected $Actn4^{-/-}$ podocyte clones are showed. The gRNA targeted region is highlighted in yellow; the PAN site is shown in light blue, and modified DNA sequence is shown in red. Podocytes maintained at permission (33°C) or differentiation (37°C) conditions for 14 d were harvested for Western blotting analysis as indicated. **(B)** The phosphorylation status of Shp2 at Y542 in these clones were also detected. **(C)** Podocytes were induced for differentiation and then maintained in attached (Att) or suspension for 30 min (Sus) before harvested for Western blotting analysis as indicated. **(D)** Podocytes were induced for differentiation and seeded on collagen type-IV–coated glass coverslips for immunofluorescence staining with anti-paxillin antibody and Hoechst for detecting FA and DNA, respectively. Scatter dot plots of the numbers of small FA (area < 1 $\mu m^2$) and matured FA (area > 1 $\mu m^2$) were shown. Data are expressed as mean ± SD from 30 representative cells in three independent experiments. Differences between continuous variables were compared using two-tail unpaired student $t$ test. ***$P$ < 0.001. ns, not statistically significant. Bars, 10 $\mu m$.
Source data are available for this figure.

clones by CRISRP/Cas9 to examine the role of ROCK2 in the generation of cellular contractility in cultured podocytes (Fig 6A). The expression of synaptopodin of these podocyte clones after temperature-switch were confirmed by Western blotting, suggesting that gene knockout of $Rock2$ did not affect the induction of podocyte differentiation in vitro. Consistent with previous results, the significant enhancement of ROCK2 activation indicated by S1366 phosphorylation and MLC phosphorylation, was observed after podocytes were induced to differentiate in vitro. Loss of ROCK2 abolished the induction of MLC phosphorylation in the differentiated podocytes (Fig 6B). The formation of FAs and stress fibers as well as interdigitating intercellular junctions were all reduced in both $Rock2^{-/-}$ clones (Fig 6C), indicating the crucial role of ROCK2 in the

maintenances of actomyosin contractility and cytoskeletal architecture in podocytes. Taken together, our data suggest that α-actinin-4 is involved in the regulation of Shp2 FA targeting to strengthen cell adhesions and cytoskeletal architecture through its effect on enhancing ROCK2-mediated cellular contractility in the cultured podocytes.

## Discussion

The intricate regulation of cell–matrix interaction and cytoskeleton contractility in response to extracellular cues is a general phenomenon affecting multiple cellular processes. In our previous

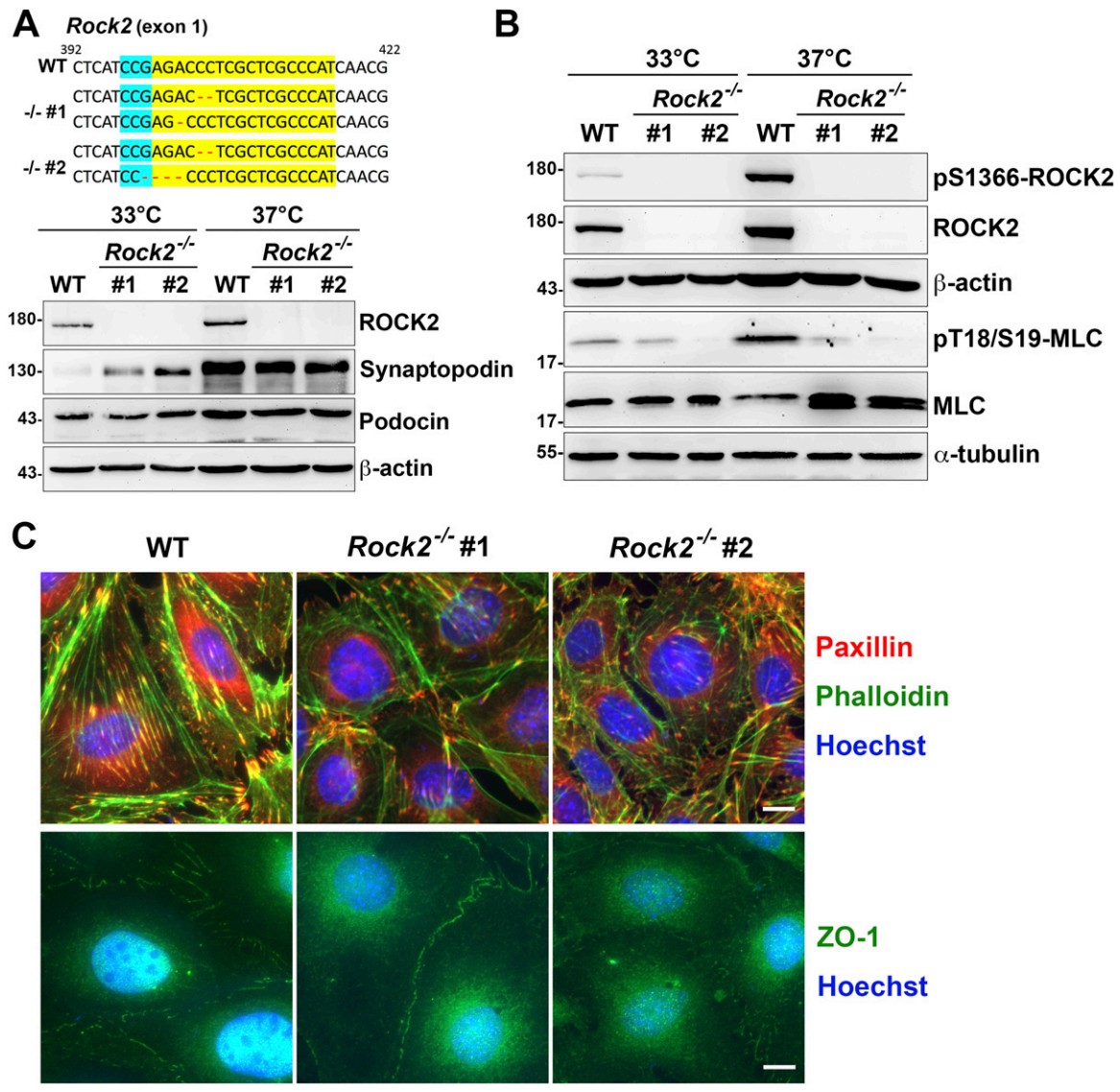

**Figure 6. The crucial role of ROCK2 in the maintenance of cellular contractility and cytoskeletal architecture in podocytes.**
**(A)** The modified DNA sequence of *Rock2* in two *Rock2*⁻/⁻ podocyte clones are showed. The gRNA targeted region is highlighted in yellow; PAN site is shown in light blue, and modified DNA sequence is shown in red. Podocytes maintained at permission (33°C) or differentiation (37°C) conditions for 14 d were harvested for Western blotting analysis as indicated. **(B)** The phosphorylation status of myosin light chain in these clones were also detected. **(C)** Podocytes were induced for differentiation and seeded on type-IV collagen–coated glass coverslips for immunofluorescence staining with anti-paxillin and anti-ZO-1 antibodies for detecting FA and intercellular junctions and phalloidin and Hoechst for F-actin and DNA, respectively. Scale bars, 10 μm.
Source data are available for this figure.

studies, we have demonstrated that the RhoA-mediated ROCK2 activation is reciprocally regulated by Src kinase and Shp2 phosphatase at FAs (Lee & Chang, 2008; Lee et al, 2010). This mechanism is essential to optimize the intracellular tension in cells response to higher matrix rigidity (Lee et al, 2013). In this study, we further demonstrated that α-actinin-4 plays a crucial role in the FA recruitment and activation of Shp2. Our data suggest that α-actinin-4 facilitates facilities Shp2 FA targeting and activation contingent upon cell adhesion signaling and contributes to the promotion of RhoA-mediated ROCK2 activation. As a result, the actomyosin contractility increased to strengthen cell adhesion. We also provided evidence showing the interplay among α-actinin-4/Shp2/ROCK2 signaling in the cultured podocytes. These findings provide a new insight of the role of α-actinin-4 in the regulation of adhesion strengthening by connecting integrin-derived signaling to ROCK-mediated actomyosin contractility, and this mechanism might be particularly critical for the maintenance of cell adhesion and architecture in podocytes.

The non-muscle α-actinins, *ACTN1* and *ACTN4*, have conserved domain structure, composed of a flexible amino-terminal F-actin–binding domain, a spectrin repeat central domain, and a carboxy-terminal calmodulin (CaM)–like domain (Sjoblom et al,

2008). They form an anti-parallel dimer and function on the formation of contractile bundles of actin filaments and on the cell adhesion maturation (Knudsen et al, 1995; Shams et al, 2012; Foley & Young, 2014; Ehrlicher et al, 2015). It has been reported that double knockdown of both *Actn1* and *Actn4* by siRNA in MEFs reduce FA maturation and force transmission in a α1-integrin–dependent manner (Roca-Cusachs et al, 2013). In retinal pigment epithelial (RPE-1) cells, double depletions of both *ACTN1* and *ACTN4* also disrupt the spatial integration of intracellular force, leading to nuclear mispositioning (Senger et al, 2019). These results suggest the involvement of non-muscle α-actinins in the processes of cell adhesion maturation and cell mechanics network connectivity. In addition, α-actinin-4 exhibits a unique mechanosensory regulation, which has not been observed in α-actinin-1 (Thomas & Robinson, 2017). Although the α-actinin-1 and α-actinin-4 have similar structure and molecular properties, the actin binding affinity of α-actinin-4 actin-binding domain is significantly lower than those of α-actinin-1 (Goldmann & Isenberg, 1993; Weins et al, 2007; Ferrer et al, 2008; Thomas & Robinson, 2017), imply that α-actinin-1 and α-actinin-4 may have different roles in the regulation of cytoskeletal dynamics under certain conditions or in some cell types.

It has been reported that α-actinin-1 is phosphorylated at Y12 residue by FAK kinase, resulting in the reduction of actin binding (Izaguirre et al, 2001). For α-actinin-4, three tyrosine residues have been reported to be phosphorylated, including Y4, Y31, and Y265 (Shao et al, 2010; Thomas & Robinson, 2017). Y4 and Y31 residues of α-actinin-4 are phosphorylated through p38 MAPK and Src kinase upon EGF stimulation, resulting in the decrease of actin binding (Shao et al, 2010; Travers et al, 2015). α-Actinin-4 can also be phosphorylated at Y265 residue; but this modification raises its affinity to actin filaments (Shao et al, 2010; Travers et al, 2013). Interestingly, the phosphorylation of α-actinin-1 at Y12 and α-actinin-4 at Y265 is critical for stress fiber establishment and FA maturation in U2OS osteosarcoma cells (Feng et al, 2013), revealing their involvement in the regulation of cytoskeleton through different mechanisms. Shp2 consists of two SH2 domains (N-SH2 and C-SH2), a PTP domain and a C-terminal tail. Shp2 keeps an auto-inhibited close conformation with intramolecular interaction between the N-SH2 and PTP domain under the basal state. Binding of a phosphotyrosine-containing peptide to the N-SH2 domain results in a conformational change to relieve this auto-inhibition (Neel et al, 2003; Song et al, 2022). In our study, we found that α-actinin-4 may either directly or indirectly interact with the Shp2 at FAs. This interaction is phosphorylation-dependent and might be controlled by FAK-mediated signaling. However, the phosphorylation site(s) on α-actinin-4 and the kinase(s) responsible for this event have not been identified. At present, our data can only suggest that α-actinin-4 positively regulated Shp2 recruitment and activation at FAs.

Podocytes exhibit a unique cytoskeletal architecture that is fundamentally linked to their function in maintaining the glomerular filtration barrier. Their cytoskeleton is thought to be contractile to maintain the cell attachment and architecture under the mechanical stress caused by blood filtration. Many attachment-associated and cytoskeleton-associated mutations related to podocyte diseases have been reported (Welsh & Saleem, 2012; Sachs & Sonnenberg, 2013; Schell & Huber, 2017). Genetic studies have highlighted the important roles of α-actinin-4 in the

maintenance of podocyte structure and attachment under mechanical stress (Dandapani et al, 2007; Weins et al, 2007). Mutations in *ACTN4* cause a wide range of alterations on α-actinin-4 properties, such as the degradation of protein stability and dynamics, which result in the impairment of cytoskeletal dynamics and the reduction of cell adhesion and spreading (Yao et al, 2004; Feng et al, 2015; Schell & Huber, 2017). It has been reported that the FSGS-associated *ACTN4* K255E mutation enhances its actin affinity and thereby impeding the protein dynamics. The mutated K255E α-actinin-4 turns into dense actin-containing aggregates and shows no association with FAs in fibroblasts (Weins et al, 2007).

It has been reported that podocyte-specific knockout of RhoA does not cause renal disease in mice (Sachs & Sonnenberg, 2013). However, expression of constitutively active RhoA, specifically in podocytes, causes proteinuria and FSGS in mice (Zhu et al, 2011); overexpression of dominant-negative RhoA also develops albuminuria and foot process effacement (Wang et al, 2012). In addition, several cytoskeletal proteins of RhoA upstream modulators are found to be involved in hereditary glomerular diseases (Schell & Huber, 2017), suggesting that inappropriate activation of RhoA is significant both physiologically and pathologically. RhoA regulates cytoskeletal rearrangements in podocytes, in response to mechanical stress, via its downstream effector ROCK (Endlich et al, 2001). Inhibition of ROCK ameliorates several experimental nephropathies (Kanda et al, 2003; Hidaka et al, 2008; Koshikawa et al, 2008). However, our data show that ROCK2 activity is indispensable for the maintenance of FAs and interdigitating intercellular junctions in cultured podocytes. Thus, RhoA/ROCK2 signaling must be tightly regulated in podocytes. In this study, we found that α-actinin-4 plays a positive role in Shp2 FA targeting and activation, which subsequently promotes RhoA-mediated ROCK2 activation for FA maturation. This α-actinin-4/Shp2/ROCK2 signaling cascade, specifically regulated at FAs, provides a mechano-transduction mechanism for the strengthening of cell adhesion.

# Materials and Methods

### Plasmids and reagents

Mouse *Actn4* cDNA was subcloned into the pmApple and pEGFP vectors. Expression construct of Shp2 were subcloned into pflag-CMV2 vector. Expression construct of mcherry-ACTN1 was obtained from JC Kuo (NYCU). Expression construct of Shp2-SWAP FRET biosensor was provided by Dr. Yingxiao Wang (UCSD) (Sun et al, 2013). Anti-paxillin antibody and mouse collagen type-IV from BD Biosciences; anti-β-actin, anti-β-tubulin, and anti-flag antibodies, FITC-phalloidin, ROCK inhibitor Y27632, and Shp2 inhibitor IIB-08 from Sigma-Aldrich; FAK inhibitor 14 from Tocris Bioscience, anti-pY397FAK, anti-pY31-paxillin, and anti-ZO-1 antibodies from Invitrogen; anti-pY542 Shp2, anti-pT18/pS19-MLC, and anti-MLC antibodies from Cell Signaling; anti-α-actinin-1 antibody from Chemicon; anti-α-actinin-4 and anti-phosphotyrosine (clone 4G10) antibodies from Millipore; anti-FAK, anti-Shp2, anti-ROCK1, and anti-ROCK2, antibodies from Santa Cruz Biotechnology Inc.; anti-synaptopodin antibody from Novus; anti-podocin antibody from

Abcam; λ protein phosphatase from New England BioLabs. Anti-pS1333 ROCK1 and anti-pS1366 ROCK2 antibodies were generated in our laboratory (Chuang et al, 2012, 2013).

## Cell culture

The wild-type and *Ptpn11*$^{Ex3-/-}$ MEFs were kindly provided by Prof. Gen-Sheng Feng (Molecular Pathology Graduate Program, UCSD) (Yu et al, 1998). MEFs and HEK293T cells were maintained in DMEM supplemented with 10% (vol/vol) of FBS in a humidified atmosphere of 5% $CO_2$/95% air at 37°C. For transient transfection experiments, cells were transfected with plasmid DNA by TurboFect transfection reagent. The conditional immortalized mouse podocyte line was kindly provided by Dr. Wen-Chih Chiang (Division of Nephrology, Department of Internal Medicine, National Taiwan University Hospital) and originally from Dr. Peter Mundel (Drug Discovery, and Biology). Podocytes, carrying a temperature-sensitive variant of the SV40 large T antigen under control of the IFNγ-inducible H-2k$^b$ promoter (Schiwek et al, 2004), were maintained in RPMI 1640 medium supplemented with 10% heat-inactivated FBS, antibiotics (10 U/ml of penicillin, 0.1 mg/ml of streptomycin, and 0.25 μg/ml of amphotericin), and 10 U/ml of IFNγ (BioLegend) in a humidified atmosphere of 5% $CO_2$/95% air at 33°C. To induce differentiation, podocytes were maintained in the medium with 2% serum in the absence of IFNγ at 37°C for 10–14 d.

## Generation of clones by CRISPR/Cas9

For single colony knockout clone selection, MEFs or podocytes were transfected with an expression construct of pZG12C01 containing Cas9 and a gRNA targeting mouse *Actn4* (GGGCCGTACTGG-TACGCCTGG) or *Rock2* (ATGGGCGAGCGAGGGTCTCGG) for 24 h. Cells were diluted and replanted for selection with 2 μg/ml of puromycin for 5–6 d. The selected single colony was amplified and the gene expression of *Actn4* or *Rock2* was checked by Western blot. The modified genomic DNA sequence of the gRNA targeting site at *Actn4* or *Rock2* gene was determined by Sanger sequencing.

## Cell treatments

For the treatment of MEFs with FAK inhibitor 14, MEFs were allowed to attach to the FN-coated dishes or glass coverslips (10 μg/ml) in the completed culture medium for 30 min and then treated with or without FAK inhibitor 14 (15 or 20 μM) for another 90 min before FA tractions isolation or PLA assays. For the treatment of podocytes with ROCK and Shp2 inhibitors, podocytes were induced to differentiation for 10 d and then replated to collagen type-IV–coated dishes (10 μg/ml) or glass coverslips (20 μg/ml) in the differentiation media for attachment overnight. Podocytes were serum-starved for 24 h and then treated with or without ROCK inhibitor Y27632 (20 μM) for 1 h or Shp2 inhibitor IIB-08 (10 or 20 μM) for 24 h in the serum-free medium.

## Isolation of focal adhesions

The FAs were isolated as described by Kuo et al (2011). Briefly, cells plated on 100-mm dishes coated with 10 μg/ml FN were washed with PBS and then hypotonic shocked with 5 ml of 2.5 mM triethanolamine in water at pH 7.0 for 2 min. Cell bodies were removed by pulsed hydrodynamic force with PBS containing 1 mM of PMSF, 5 mM of NaF, and 0.2 mM of $Na_3VO_4$ using Waterpik (setting "3.5," Interplak dental water jet WJ6RW; Conair). The FAs remained on dishes were collected by scraping with a rubber policeman with 2× Laemmli buffer for Western blot analysis. For in vitro pull-down analysis, the FAs were harvested with a buffer (0.5% Triton-X100, 50 mM Tris–HCl, pH 8.0, 150 mM NaCl, 0.1% DOC, 1 mM PMSF, 50 mM NaF, 2 mM $Na_3VO_4$, and protease and phosphatase inhibitor cocktails). The precleared supernatants were incubated with 50 μg of His-tagged N-SH2 recombinant protein for 20 min, followed by pull-down with Ni-beads for 40 min at 4°C. After extensive washing, the pulled-down materials were separated by SDS–PAGE and detected by silver staining or Western blotting analysis. For immunoprecipitation, FAs were harvested with the IP buffer (0.5% Triton-X100, 25 mM Tris–HCl, pH 7.4, 50 mM NaCl, 0.1% SDS, 0.5% DOC, 1 mM PMSF, 50 mM NaF, 2 mM $Na_3VO_4$, and protease and phosphatase inhibitor cocktails). The precleared supernatants were then immunoprecipitated with anti-Shp2 antibody (B-1) or normal IgG as control.

## Protein identification by LC/MS/MS

The gel samples were digested with modified trypsin, chymotrypsin, or Asp-N (multiple enzymes used to increase sequence coverage), and the peptides were extracted with 0.1% formic acid and dried in a Speed-Vac. The sample was resuspended in 0.1% formic acid immediately before MS analysis. A Thermo Finnigan LTQ Orbitrap tandem mass spectrometer interfaced with an Agilent 1100D HPLC system (Proteomes Research Center, NYCU) was used for electrospray ionization–ion trap tandem mass spectrometry.

## Shp2 activity assay by a FRET biosensor

MEFs were seeded on FN-coated glass bottom dishes (10 μg/ml) and transfected with Shp2-SWAP for imaging (Sun et al, 2013). During imaging, cells were maintained in phenol red–free DMEM containing 10% FBS and 10 mM of Hepes buffer at 37°C. Images were obtained on an inverted microscope (Nikon Ti-E) equipped with a CCD digital camera (Hamamatsu ORCA-ER). The following filter sets (Chroma) were used in our experiments for FRET imaging: a dichroic mirror (458 nm), an excitation filter 438/24 nm, an ECFP emission filter 483/32 nm, and a FRET emission filter 542/27 nm. The pixel-by-pixel images of FRET efficiency were obtained based on the background-subtracted fluorescence intensity image of ECFP and FRET by using NIS-AR software (Nikon).

## Immunofluorescence staining

Cells were fixed with 4% paraformaldehyde in phosphate-buffered saline, pH 7.4, for 30 min, followed by permeabilization with Tris-buffered saline containing 0.3% Triton X-100 for 5 min. After blocking with 5.5% normal goat serum for 30 min, samples were incubated overnight at 4°C with antibodies, then for 1 h with Alexa488- or Alexa568-conjugated secondary antibodies, phalloidin, and Hoechst, washed, mounted, and examined on a fluorescence microscope (Nikon Ti-E), equipped with a 60× oil immersion

objective lens. Images were captured using a cooled CCD digital camera (Hamamatsu ORCA-ER), with the associated imaging software, and arranged using Photoshop (Adobe) software. For the quantification of FA size, the images of IF staining with anti-paxillin antibody were segmented and analyzed using ImageJ software.

## In situ PLA

The PLA assay was carried out by the Duolink kit. Briefly, cells were fixed, permeabilized, and incubated with two primary antibodies in different species, followed by two DNA strand–attached second antibodies. When two second antibodies probes interacted, the rolling circle-forming DNA connector oligonucleotides and ligase were added, and PCR reaction was coupled, followed by the addition of the fluorescence probe for subsequent detection using a fluorescence microscope as described above.

## Supplementary Information

## Acknowledgements

We thank Dr. Wen-Chih Chiang (National Taiwan University College of Medicine) for the mouse podocyte line (originally from Dr. Peter Mundel's laboratory); and Dr. Yinxiao Wang (Institute of Engineering in Medicine, UCSD) for providing the plasmid of the FRET Shp2 biosensor. This work was financially supported by the Ministry of Science and Technology in Taiwan (MOST105-2628-B-010-009-MY3 and MOST109-2314-B-010-054-MY3) and by the Center for Intelligent Drug Systems and Smart Bio-devices (IDS2B), National Yang Ming Chiao Tung University from the Featured Areas Research Center Program within the framework of the Higher Education Sprout Project by the Ministry of Education in Taiwan.

### Author Contributions

C-C Tseng: data curation, formal analysis, validation, and investigation.
R-H Zheng: data curation, formal analysis, validation, and investigation.
T-W Lin: data curation, formal analysis, and investigation.
C-C Chou: formal analysis and investigation.
Y-C Shih: formal analysis and investigation.
S-W Liang: investigation.
H-H Lee: conceptualization, funding acquisition, investigation, project administration, and writing—original draft, review, and editing.

### Conflict of Interest Statement

The authors declare that they have no conflict of interest.

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
