## [Reviewer comments · Life Science Alliance]

Life Science Alliance

Alpha-actinin-4 recruits Shp2 into focal adhesions to potentiate ROCK2 activation in podocytes

Chien-Chun Tseng, Ru-Hsuan Zheng, Ting-Wei Lin, Chih-Chiang Chou, Yu-Chia Shih, and Hsiao-Hui Lee
DOI: <https://doi.org/10.26508/lsa.202201557>

Corresponding author(s): Hsiao-Hui Lee, National Yang Ming Chiao Tung University

Review Timeline:

Submission Date:	2022-06-10
Editorial Decision:	2022-06-10
Revision Received:	2022-08-18
Editorial Decision:	2022-08-19
Revision Received:	2022-08-20
Accepted:	2022-08-22

Transaction Report:

Please note that the manuscript was previously reviewed at another journal and the reports were taken into account in the decision-making process at Life Science Alliance.

Comments to the Authors (Required):

This manuscript by Lee and co-workers extends their previous studies on the role of SHP2 in regulation of focal adhesions. Here they identify ACTN4 as a protein that specifically recruits SHP2 to focal adhesions where SHP2 functions to dephosphorylate ROCK2 allowing for enhanced Rho effector functions including increased stress fibers and focal adhesions. Using a podocyte cell line, they validate their findings in a model involving differentiation of podocytes in vitro. Differentiation appears to function through a mechanism of increased adhesion that involves SHP2. These changes are dependent on ACTN4 and recruitment of SHP2.

In general, the findings are of great interest and contribute to a better understanding of SHP2 in integrin function as well as extending these findings to the function of glomerular podocytes. However, the data has significant gaps which require further study.

1. In Figure 1B, the authors claim that they identify a single protein that appears to co-precipitate with the N-terminal SH2 domain of SHP2 that is lost after FAK inhibition. It seems a bit astounding that this is the only band that is different. For example, there is another band around 38 kD that is also changed. Some more information about this experiment would be helpful including what other molecules were identified.
2. The specificity of binding is argued to be ACTN4 specific as no ACTN1 was co-precipitated. This experiment is lacking many controls like the sensitivity of the ACTN1 antibody and demonstration that similar or greater amount of ACTN1 is present in these cells. What about ACTN2 or ACTN3? The binding to the N-terminal SH2 domain suggests that this is a phosphotyrosine dependent interaction. Is the ACTN4 tyrosine phosphorylation? Does the binding go away with phosphatase treatment? Could the specificity be due to differences in tyrosine phosphorylation?
3. Similarly, the FRET experiments suggest that SHP2 binding to ACTN4 is activating. What is the mechanism of this activation event? Assuming that ACTN4 is phosphorylated, what is the kinase, a SRC kinase?
4. Podocyte differentiation is a somewhat artificial system based on IFN induced expression of a temperature sensitive polyoma middle T antigen. Why removal of this oncogene allows for differentiation is not clear. The authors spend some time validating the differentiation conditions. The immunoblots however lack molecular weight markers and controls. It is controversial, for example, whether these podocyte cell lines express nephrin, so molecular weight markers are essential here. What the data do show here is clear that both SHP2 and ACTN4 are important for attainment of the "differentiated" phenotype at least as defined by adhesion but not by synaptopodin expression. The worry here is that this is just a correlation. Loss of ACTN4 likely impairs

FA and defects in Rho activation are expected. Is the SHP2 defect an epiphenomenon or directly correlated to the absence of ACTN4?

Reviewer #2 Review

Comments to the Authors (Required):

The manuscript from Chien-Chun Tseng and colleagues aimed to further investigate the regulation of the phosphatase SHP2 (encoded by PTPN11) at focal adhesions. Using mouse embryonic fibroblasts, they report the localisation of SHP2 to focal adhesions and an interaction with alpha-actinin-4. In immortalised mouse podocytes the authors report an increase in SHP2 following a switch from 33oC to 37oC and an associated increase in FAK and paxillin phosphorylation, ROCK activation and myosin light chain (MLC) phosphorylation. Continuing in podocytes the authors next show that the Rho compound Y27632 reduced ROCK activation and MLC phosphorylation increased cortical actin and altered cell junctions. Furthermore, the SHP2 inhibitor IIB-08 increased albumin permeability measured with a BSA filtration assay. In Actn4 CRISPR-knockout podocytes there was no SHP2-alpha-actinin-4 signal using a proximity ligation assay and there was reduced SHP2 phosphorylation. The authors concluded by examining ROCK2 in podocytes and report that ROCK2 knockout abolished MLC phosphorylation. The authors summarise that alpha-actinin-4 interacts with SHP2 and regulates SHP2 focal adhesion targeting via ROCK2 mediated contractility.

In previous publications the authors have focussed on the role of SHP2 at focal adhesions. The advance in this paper appears to be the interaction between SHP2 and alpha-actinin-4 and the biochemical data from MEFs appears convincing. By contrast the podocyte data is less convincing and the statement "this molecular mechanism is crucial for the maintenance of the filtration function of podocytes under mechanical stress" is not supported by the data presented.

I have the following comments, questions, and suggestions for the authors:

Figure 1: A: The reduction in Shp2 is minimal. Was this replicated? Did it change with length of MEF exposure to fibronectin? Molecular weight markers are absent. Please provide the whole membrane for the pull-down assay. The PLA data appear convincing. How many replicates were performed?

Figure 2: B: Why isn't the FRET efficiency abolished if there is no alpha-actinin-4 in the MEFs? Is this background signal? What is the efficiency of the m-apple alpha-actinin-4 transfection?

Figure 3: The detection of podocyte markers in immortalised podocytes is notoriously low (PMID: 21632959). Indeed, it is well recognised that podocytes in culture are a limited representation of podocytes in vivo. They do not form podocyte junctions (slit diaphragms) or have convincing primary and tertiary foot processes. The authors do not state these limitations. In this figure it is therefore surprising in this system to see nephrin and podocin at all and certainly at 33oC when the cells are proliferating. Western blots need molecular weights.

Figure 4. B. The differentiated podocytes do not typically form a complete monolayer and as such BSA assays are challenging due to the wide variation. How did the authors overcome this issue? Were the cells over confluent? Images of the cells during/after the filtration assay would be useful here. This albumin permeability assay is a poor measure of podocyte filtration function in vivo and I suggest removing the association between these findings and podocyte filtration function. To test podocyte filtration function an in vivo system is necessary- with zebrafish or in vivo studies in mice.

Figure 5: B Molecular weight markers are absent again.

Minor comments

1. Nomenclature SHP-2 is encoded by the PTPN11. It is a protein tyrosine phosphatase. This introduction to terminology would be useful so as not to confuse with the lipid phosphatase SHIP1/SHIP2 which are also linked to podocyte function.
2. On page 4 the statement "podocyte is a kind of glomerular-specific epithelial cells that cover the outer surfaces of the glomerular capillaries by their interdigitating foot processes (FPs) for blood ultrafiltration [27-29]" is vague and colloquial.
3. There are many grammar and spelling errors.

Reviewer #3 Review

Comments to the Authors (Required):

There is a large body of work demonstrating that actinin 4 is associated with podocyte abnormalities. To this end this protein has been deleted in podocytes with a phenotype (J Biol Chem282:467-477). There have also been mutant mice made with changes in actinin-4 that cause FSGS (PNAS 2007; vol 104 page 16080)and it has been shown that S159 phosphorylation of S159 leads to pathology in podocytes. This manuscript makes a small advance as it proposes that Shp2 is recruited to FAs by interacting a-

actinin-4. This leads to Rho A mediated activation required for FAs which increases the strength of cell adhesion.

I have some major comments about the manuscript:

- 1) The manuscript shows that there is an interaction with Shp2 and actinin 4. Is this direct or indirect. The figure 3A implies specific interactions between the components in the diagram. There is no evidence that this is actually how it fits together. At least some of this work should be done for the work to be significant.
2. There is no data to substantiate the claim that actinin-4 plays a role in Shp2 targeting to focal adhesions.
3. As of now the manuscript has correlative data linking Shp2 and actinin 4 and Rock. In addition the data linking this to integrins is very weak.

As such I believe this paper is simply descriptive and correlative right now and in depth studies need to be done to back up the claims made in the manuscript.

June 10, 2022

Re: Life Science Alliance manuscript #LSA-2022-01557-T

Prof. Hsiao-Hui Lee
National Yang Ming University
Department of Life Sciences and Institute of Genome Sciences
No. 155, Sec. 2, Linong Street
Taipei 11221
Taiwan

Dear Dr. Lee,

Thank you for submitting your manuscript entitled "Alpha-actinin-4 recruits Shp2 into focal adhesions to potentiate ROCK2 activation in podocytes" to Life Science Alliance. We invite you to submit a revised manuscript addressing the following Reviewer comments:

- Address Reviewer 1's Points #1, 2 and 4
- Address Reviewer 2's comments, with the removal of the filtration assay, and toning down of the associated conclusions
- Tone down conclusions related to the comments outlined by Reviewer 3

Thank you for this interesting contribution to Life Science Alliance. We are looking forward to receiving your revised manuscript.

Sincerely,

- A letter addressing the reviewers' comments point by point.
- An editable version of the final text (.DOC or .DOCX) is needed for copyediting (no PDFs).
- High-resolution figure, supplementary figure and video files uploaded as individual files: See our detailed guidelines for preparing your production-ready images, <https://www.life-science-alliance.org/authors>
- Summary blurb (enter in submission system): A short text summarizing in a single sentence the study (max. 200 characters including spaces). This text is used in conjunction with the titles of papers, hence should be informative and complementary to the title and running title. It should describe the context and significance of the findings for a general readership; it should be written in the present tense and refer to the work in the third person. Author names should not be mentioned.
- By submitting a revision, you attest that you are aware of our payment policies found here: <https://www.life-science-alliance.org/copyright-license-fee>

B. MANUSCRIPT ORGANIZATION AND FORMATTING:

Responses to the Reviewers

Reviewer #1:

This manuscript by Lee and co-workers extends their previous studies on the role of SHP2 in regulation of focal adhesions. Here they identify ACTN4 as a protein that specifically recruits SHP2 to focal adhesions where SHP2 functions to dephosphorylate ROCK2 allowing for enhanced Rho effector functions including increased stress fibers and focal adhesions. Using a podocyte cell line, they validate their findings in a model involving differentiation of podocytes in vitro. Differentiation appears to function through a mechanism of increased adhesion that involves SHP2. These changes are dependent on ACTN4 and recruitment of SHP2.

In general, the findings are of great interest and contribute to a better understanding of SHP2 in integrin function as well as extending these findings to the function of glomerular podocytes. However, the data has significant gaps which require further study.

1. In Figure 1B, the authors claim that they identify a single protein that appears to co-precipitate with the N-terminal SH2 domain of SHP2 that is lost after FAK inhibition. It seems a bit astounding that this is the only band that is different. For example, there is another band around 38 kD that is also changed. Some more information about this experiment would be helpful including what other molecules were identified.

Response:

The mass results of the FAK-sensitive N-SH2-pull-down proteins was added as supplemental Table S1. Data information including: gene and protein name, calculated molecular weight, the number of identified spectra, and the amino acid coverage percentage of identified peptide sequence. Among the identified proteins, α -actinin-4, α -actinin-1, and vimentin are significant and representative.

2. The specificity of binding is argued to be ACTN4 specific as no ACTN1 was co-precipitated. This experiment is lacking many controls like the sensitivity of the ACTN1 antibody and demonstration that similar or greater amount of ACTN1 is present in these cells. What about ACTN2 or ACTN3? The binding to the N-terminal SH2 domain suggests that this is a phosphotyrosine dependent interaction. Is the ACTN4 tyrosine phosphorylation? Does the binding go away with phosphatase treatment? Could the specificity be due to differences in tyrosine phosphorylation?

Response:

- (1) As suggested by the reviewer, we replaced this immunoblots by two sets of data results showing that α -actinin-4 was co-precipitated with Shp2 (Fig 1C), while α -actinin-1 did not (Fig 1D). The sensitivity of anti-ACTN1 Ab was also confirmed (F1D). We also added a supplemental data showing the co-precipitation of exogenous expressed flag-Shp2 with GFP-ACTN4, but not mCherry-ACTN1 in HEK-293T cells (Figs S2A and B), suggesting the specificity of the interaction between α -actinin-4 and Shp2.
 - (2) Since ACTN2 and ACTN3 are expressed specifically in muscle cells, we did not study these two isoforms in this study.
 - (3) We added two experiments to clarify the tyrosine phosphorylation of α -actinin-4. By IP/WB (anti-pY; 4G10), we observed that GFP- α -actinin-4 was tyrosine phosphorylated (Fig S2C). By using λ PPase to pre-treat FA fraction, we demonstrated that the interaction of α -actinin-4 with Shp2 N-SH2 is phosphorylation dependent (Fig 1E).
3. Similarly, the FRET experiments suggest that SHP2 binding to ACTN4 is activating. What is the mechanism of this activation event? Assuming that ACTN4 is phosphorylated, what is the kinase, a SRC kinase?

Response:

- (1) Shp2 consists of two SH2 domains (N-SH2 and C-SH2), a PTP domain, and a C-terminal tail (C-tail) containing two major tyrosine phosphorylation sites (Y542 and Y580). Under basal state, Shp2 keeps an auto-inhibited close conformation with intramolecular interaction between the N-SH2 and PTP domain. Binding of a phosphotyrosine-containing peptide to the N-SH2 domain may result in a conformational change that relieves this autoinhibition and becomes activated. Phosphorylation of Shp2 at C-tail also results in the activation through its intramolecular interaction with N-SH2 domain [Neel et al., Trends Biochem Sci, 2003 and Song et al., Med Res Rev, 2022]. Therefore, we speculate that α -actinin-4 has the potential to act as Shp2 activator or adaptor at FAs, facilitating Shp2 activation by binding to N-SH2 domain or mediating Shp2 C-tail phosphorylation.
- (2) It has been reported that α -actinin-4 could be phosphorylated at three tyrosine residues (Y4, Y31, and Y265) [Thomas et al., Semin Cell Dev Biol, 2017]. However, the phosphorylation site(s) on α -actinin-4 and the kinase(s) responsible for this event have not been identified. At present, our data can only suggest the involvement of α -actinin-4 in the regulation of Shp2 recruitment and activation at FAs. We added a paragraph about the tyrosine phosphorylation of α -actinin-4 and in the Discussion section (page 11).

4. Podocyte differentiation is a somewhat artificial system based on IFN induced expression of a temperature sensitive polyoma middle T antigen. Why removal of this oncogene allows for differentiation is not clear. The authors spend some time validating the differentiation conditions. The immunoblots however lack molecular weight markers and controls. It is controversial, for example, whether these podocyte cell lines express nephrin, so molecular weight markers are essential here. What the data do show here is clear that both SHP2 and ACTN4 are important for attainment of the "differentiated" phenotype at least as defined by adhesion but not by synaptopodin expression. The worry here is that this is just a correlation. Loss of ACTN4 likely impairs FA and defects in Rho activation are expected. Is the SHP2 defect an epiphenomenon or directly correlated to the absence of ACTN4?

Response:

(1) We thank the reviewer for this important comment. As suggested by the reviewer, we added molecular weight (MW) markers on all WB data in the study. The calculated MW of synaptopodin renal isoform is 96 kDa, and the Synaptopodin bands were detected at approximately 130kDa in our immunoblots. We have tried three different anti-synaptopodin antibodies and got similar results (Fig R1). It is consistent with other reports showing a higher mass than expected, possibly due to post-translational modification [Ning et al., JASN, 2020].

[Figure removed by LSA Editorial Staff per authors' request]

(2) The predicted MW of nephrin is about 138kDa. However, the major observed band size in the blots using anti-nephrin antibody (Abcam ab58968) was approximately 100kDa (as described in the datasheet). We tried other two different anti-nephrin antibodies (Santa Cruz and Abcam ab216341), but we got inconsistent results. Therefore, we removed the WB data for nephrin. The predicted MW of podocin is 42 kDa, and the major band size detected in our WB data for podocin data was also approximately 42kDa. Two additional bands with smaller MW were also observed in our data, similar to the description in the antibody datasheet (Abcam ab50339). Therefore, we removed

all the WB blots for nephrin and instead of the WB blots for podocin (Fig 4A, 5A, and 6A).

- (3) We agree with the reviewer's concern that our current data are insufficient to support a causal relationship between loss of ACTN4 and Shp2 deficiency. In our data, loss of ACTN4 did not affect cell adhesion, spreading, and the FA formation. The phosphorylation of FAK Y397 was also unchanged (Fig R2), suggesting the basal activity of FA might be similar. We observed that Shp2 activity was significantly diminished, and a decrease of the number of mature FA ($>1\mu\text{m}^2$) in ACTN4^{-/-} cells. Since we cannot rule out the possibility that ACTN4 indirectly regulates Shp2 through other factors in FAs, we remove the proposed model, which showed a molecular link between ACTN4 and Shp2 (Fig 3A in original version). Instead, we emphasize that ACTN4 plays a positive and important role for Shp2 signaling in FAs.

[Figure removed by LSA Editorial Staff per authors' request]

Reviewer #2

The manuscript from Chien-Chun Tseng and colleagues aimed to further investigate the regulation of the phosphatase SHP2 (encoded by PTPN11) at focal adhesions. Using mouse embryonic fibroblasts, they report the localisation of SHP2 to focal adhesions and an interaction with alpha-actinin-4. In immortalised mouse podocytes the authors report an increase in SHP2 following a switch from 33oC to 37oC and an associated increase in FAK and paxillin phosphorylation, ROCK activation and myosin light chain (MLC) phosphorylation. Continuing in podocytes the authors next show that the Rho compound Y27632 reduced ROCK activation and MLC phosphorylation increased cortical actin and altered cell junctions. Furthermore, the SHP2 inhibitor IIB-08 increased albumin permeability measured with a BSA filtration assay. In Actn4 CRISPR-knockout podocytes there was no SHP2-alpha-actinin-4 signal using a proximity ligation assay and there was reduced SHP2 phosphorylation. The authors concluded by examining ROCK2 in podocytes and report that ROCK2 knockout abolished MLC phosphorylation. The authors summarise that alpha-actinin-4 interacts with SHP2 and regulates SHP2 focal adhesion targeting via ROCK2 mediated contractility.

In previous publications the authors have focussed on the role of SHP2 at focal adhesions. The advance in this paper appears to be the interaction between SHP2 and alpha-actinin-4 and the biochemical data from MEFs appears convincing. By contrast the podocyte data is less convincing and the statement "this molecular mechanism is crucial for the maintenance of the filtration function of podocytes under mechanical stress" is not supported by the data presented. I have the following comments, questions, and suggestions for the authors:

1. Figure 1: A: The reduction in Shp2 is minimal. Was this replicated? Did it change with length of MEF exposure to fibronectin? Molecular weight markers are absent. Please provide the whole membrane for the pull-down assay. The PLA data appear convincing. How many replicates were performed?

Response:

- (1) The experiment of FA fractionation was replicated (Fig S1), and the statistical results of the relative FAK Y397 phosphorylation and levels of Shp2 versus vinculin in FA fractions were measured and showed (Fig 1A).
- (2) Based on our previous experience of FA isolation experiment, Shp2 was consistently detectable in the FA fraction (although only ~1% of total Shp2) regardless of whether cells were attached to FN for 2 or 4 hours or longer. Considering the efficacy of the treatment of FAK inhibitor and controlling of cell density, we maintained 2-hours incubation in all the experiment with cells treated with or without FAK inhibitor.
- (3) The molecular weight markers of all the WB data were added. The original data of WB results were provided as supplemental data (source data).
- (4) The PLA experiment was replicated three times with consistent results (Fig2A). The information about these data analysis was added in the figure legends.

2. Figure 2: B: Why isn't the FRET efficiency abolished if there is no alpha-actinin-4 in the MEFs? Is this background signal? What is the efficiency of the m-apple alpha-actinin-4 transfection?

Response:

- (1) Yes, it is very likely as the background signal. We added a supplemental data showing the FRET efficiency of Shp2-SWAP in WT cells on FN vs. on poly-L-lysine (PLL) (Fig S4). The FRET efficiency in cells on PLL could be presented as the background signal (without integrin activation). Thus, the low FRET efficiency detected in ACTN4^{-/-} cells in suggests the importance of ACTN4 for integrin-mediated Shp2 activation.
- (2) The transfection efficiency of Shp2-SWAP was about 5% in MEFs, while the co-

transfection efficiency of Shp2-SWAP and mApple-ACTN4 was lower.

3. Figure 3: The detection of podocyte markers in immortalised podocytes is notoriously low (PMID: 21632959). Indeed, it is well recognised that podocytes in culture are a limited representation of podocytes in vivo. They do not form podocyte junctions (slit diaphragms) or have convincing primary and tertiary foot processes. The authors do not state these limitations. In this figure it is therefore surprising in this system to see nephrin and podocin at all and certainly at 33°C when the cells are proliferating. Western blots need molecular weights.

Response:

- (1) We agree with reviewer's opinions and added the following sentence "Although cultured podocytes do not have convincingly convincing primary and tertiary foot processes to form the podocyte specific slit diaphragms, the significant activation of Shp2 and ROCK2 detected in the cells after temperature-switch suggests that it could be a feasible cell model to verify our hypothesis." in the revised version (page 8).
 - (2) We checked the WB data for nephrin and podocin. The predicted MW of nephrin is about 138kDa. However, the major observed band size in the blots was approximately 100kDa (similar to the results on the antibody datasheet Abcam ab58968). We tried other two anti-nephrin antibodies (Santa Cruz and Abcam ab216341), but got un-consistent results. Therefore, we removed all the WB data for nephrin. The predicted MW of podocin is 42 kDa, and the major band size detected in our WB data for podocin data was also approximately 42kDa. Two additional bands with smaller MW were also observed in our data, similar to the description in the antibody datasheet (Abcam ab50339). And the data showed that expression of podocin was expressed in the podocytes under both 33°C and 37°C conditions. We added the WB data for podocin in Fig 5A and Fig6A of revised version.
 - (3) The molecular weight markers of all WB data were added.
4. Figure 4. B. The differentiated podocytes do not typically form a complete monolayer and as such BSA assays are challenging due to the wide variation. How did the authors overcome this issue? Were the cells over confluent? Images of the cells during/after the filtration assay would be useful here. This albumin permeability assay is a poor measure of podocyte filtration function in vivo and I suggest removing the association between these findings and podocyte filtration function. To test podocyte filtration function an in vivo system is necessary- with zebrafish or in vivo studies in mice.

Response:

We agree with the reviewer on this point. Although we did control the cell density (~5 X10⁴ cells/cm²) on the beginning of the experiment, we did not have proper evidence to support the status of cell confluence. Therefore, we have removed this data and the descriptions about the functional examination of podocyte filtration in the revised manuscript.

5. Figure 5: B Molecular weight markers are absent again.

Response:

The molecular weight markers of all WB data were added.

Minor comments

1. Nomenclature SHP-2 is encoded by the PTPN11. It is a protein tyrosine phosphatase. This introduction to terminology would be useful so as not to confuse with the lipid phosphatase SHIP1/SHIP2 which are also linked to podocyte function.

Response:

As suggested by the reviewer, we added this description in the Introduction (page 3).

2. On page 4 the statement "podocyte is a kind of glomerular-specific epithelial cells that cover the outer surfaces of the glomerular capillaries by their interdigitating foot processes (FPs) for blood ultrafiltration [27-29]" is vague and colloquial.

Response:

We have clarified this sentence in the revised manuscript, which now reads:

"Podocytes are highly specialized visceral epithelial cells lining the outer surface of the glomerular capillaries. They have a complex cellular architecture consisting of cell body and major processes (MPs) that extend outward from their cell body, forming interdigitated foot processes (FPs) that enwrap the glomerular capillaries [29, 30]." (page 4)

3. There are many grammar and spelling errors.

Response:

Grammar and spelling errors have been carefully corrected.

August 19, 2022

RE: Life Science Alliance Manuscript #LSA-2022-01557-TR

Prof. Hsiao-Hui Lee
National Yang Ming University
Department of Life Sciences and Institute of Genome Sciences
No. 155, Sec. 2, Linong Street
Taipei 11221
Taiwan

Dear Dr. Lee,

Thank you for submitting your revised manuscript entitled "Alpha-actinin-4 recruits Shp2 into focal adhesions to potentiate ROCK2 activation in podocytes". We would be happy to publish your paper in Life Science Alliance pending final revisions necessary to meet our formatting guidelines.

- please add the Twitter handle of your host institute/organization as well as your own or/and one of the authors in our system
- please use the [10 author names, et al.] format in your references (i.e. limit the author names to the first 10)
- please fix the typo in your Figure 2 legend (you have a panel G rather than panel B)

A. FINAL FILES:

B. MANUSCRIPT ORGANIZATION AND FORMATTING:

Sincerely,

August 22, 2022

RE: Life Science Alliance Manuscript #LSA-2022-01557-TRR

Prof. Hsiao-Hui Lee
National Yang Ming Chiao Tung University
Department of Life Sciences and Institute of Genome Sciences
No. 155, Sec. 2, Linong Street
Taipei 11221
Taiwan

Dear Dr. Lee,

Thank you for submitting your Research Article entitled "Alpha-actinin-4 recruits Shp2 into focal adhesions to potentiate ROCK2 activation in podocytes". It is a pleasure to let you know that your manuscript is now accepted for publication in Life Science Alliance. Congratulations on this interesting work.

DISTRIBUTION OF MATERIALS:

Again, congratulations on a very nice paper. I hope you found the review process to be constructive and are pleased with how the manuscript was handled editorially. We look forward to future exciting submissions from your lab.

Sincerely,
